# A Potential Role of Semaphorin 3A during Orthodontic Tooth Movement

**DOI:** 10.3390/ijms22158297

**Published:** 2021-08-02

**Authors:** Sinan Şen, Christopher J. Lux, Ralf Erber

**Affiliations:** Department of Orthodontics and Dentofacial Orthopaedics, University of Heidelberg, Im Neuenheimer Feld 400, 69120 Heidelberg, Germany; christopher.lux@med.uni-heidelberg.de (C.J.L.); ralf.erber@med.uni-heidelberg.de (R.E.)

**Keywords:** tooth movement, mechanotransduction, neuronal guidance molecules, Semaphorins, bone remodeling

## Abstract

Background: Induced tooth movement during orthodontic therapy requires mechano-induced bone remodeling. Besides various cytokines and growth-factors, neuronal guidance molecules gained attention for their roles in bone homeostasis and thus, potential roles during tooth movement. Several neuronal guidance molecules have been implicated in the regulation of bone remodeling. Amongst them, Semaphorin 3A is particular interesting as it concurrently induces osteoblast differentiation and disturbs osteoclast differentiation. Methods: Mechano-regulation of Sema3A and its receptors PlexinA1 and Neuropilin (RT-qPCR, WB) was evaluated by applying compressive and tension forces to primary human periodontal fibroblasts (hPDLF) and alveolar bone osteoblasts (hOB). The association of the transcription factor Osterix (SP7) and SEMA3A was studied by RT-qPCR. Mechanisms involved in SEMA3A-mediated osteoblast differentiation were assessed by Rac1GTPase pull-downs, β-catenin expression analyses (RT-qPCR) and nuclear translocation assays (IF). Osteogenic markers were analyzed by RT-qPCR. Results: SEMA3A, PLXNA1 and NRP1 were differentially regulated by tension or compressive forces in hPDLF. Osterix (SP7) displayed the same pattern of regulation. Recombinant Sema3A induced the activation of Rac1GTPase, the nuclear translocation of β-catenin and the expression of osteogenic marker genes. Conclusion: Sema3A, its receptors and Osterix are regulated by mechanical forces in hPDLF. SEMA3A upregulation was associated with Osterix (SP7) modulation. Sema3A-enhanced osteogenic marker gene expression in hOB might be dependent on a pathway involving Rac1GTPase and β-catenin. Thus, Semaphorin 3A might contribute to bone remodeling during induced tooth movement.

## 1. Introduction

Tooth and jaw misalignments with treatment needs occur with a high prevalence [1,2]. Although generally well tolerated, there are side effects during orthodontic therapy with fixed appliances. In addition to temporary pain perception by the patient, these include a higher risk of gingivitis and demineralization due to the limited possibilities of oral hygiene during treatment with fixed appliances [3]. However, by far the most serious side effect is the possibility of the development of orthodontically induced root resorptions (OIRR) [4]. Such root resorptions are likely closely linked to resorptive events of the bone during orthodontic tooth movement, also at the molecular level [5,6]. Therefore, accurate and comprehensive knowledge of the biology of tooth movement would be key to understanding and possibly minimizing the occurrence of such side effects.

The biology of orthodontic tooth movement to correct tooth misalignments has gained importance in recent years [7]. The general aim is to identify molecular factors activated by the application of force. During orthodontic tooth movement, the action of orthodontic forces leads to mechanically induced changes in the alveolar bone. The periodontal ligament (PDL) lying in the direction of movement is compressed (“pressure side”), while the PDL lying against the tooth movement is stretched (“tension side”). As a result, bone resorption predominates on the pressure side and bone apposition on the tension side and enables the tooth to move within the alveolar bone. Although bone cells of every anatomical location (including osteoblasts, osteoclasts and osteocytes) are sensitive to mechanical loads and thus adapt to altered external conditions [8,9], it is now recognized that, for orthodontic tooth movement, the force acting on the periodontal ligament (PDL) and fibroblasts localized within is decisive. However, the exact modes of force reception and subsequent mechanotransduction are not fully understood. Neither have the molecular factors involved in this process been identified and functionally characterized completely.

Apart from their primary roles in neuronal development, functions of neural guidance molecules in bone homeostasis and remodeling have been described. Members from the Ephrin/Eph receptor family [10,11], as well as Semaphorins and their receptors Neuropilin and Plexin [12] have so far been implicated in maintaining bone homeostasis. The involvement of Ephrines and Eph receptors in bone remodeling during orthodontic tooth movement has been studied in our laboratory [13,14]. To date, no involvement of Semaphorins during orthodontic tooth movement has been demonstrated. However, Semaphorin 3A (Sema3A) would be of particular interest in this role as it concurrently induces osteoblast differentiation and disturbs osteoclast differentiation [12]. Hayshi et al. studied Semaphorin 3A functions in a mouse model and showed that hematopoietic osteoclast progenitor cells co-cultured with osteoblasts differentiated into osteoclasts only when the osteoblasts were Sema3A-deficient. Exogenous Sema3A was able to prevent osteoclastogenic differentiation in macrophages from the bone marrow but only if Sema3A stimulation took place before the addition of Rankl. The Sema3A effect was also dependent on the expression of Nrp1 on osteoclasts. Interestingly, Sema3A itself was responsible for the induction of Nrp1; Rankl, on the other hand, was able to prevent this induction via an NF-κB-dependent signaling pathway. Both results indicate that Nrp1 is the decisive co-receptor for the Sema3A-dependent regulation of osteoblasts and osteoclasts. However, the actual signal transmission takes place via the Plexin-A1 receptor (PlxnA1). Nrp1 competes with TREM-2 for the binding of Plxna1, so that high Nrp1 abundance, as caused by Sema3A, prevents TREM-2 activation and thereby the NFATc1-dependent differentiation of progenitor cells into osteoclasts. In Sema3A-deficient animals, osteoblast formation and differentiation are reduced. In vitro, it could be shown that, after Sema3A stimulation, PlxnA1 interacts with FARP-2 and Rac1GTPase is activated. This led to the nuclear translocation of the Wnt-dependent transcriptional co-regulator β-catenin and to the induction of osteogenic differentiation. When analyzing the mechanical regulation of Ephrin-A2 and Ephrin-B2 in hPDLF during tooth movement, we were able to identify a putative signal transduction pathway. The identification of signaling factors opens the possibility of influencing tooth movement by targeting signaling molecules and pathways. For this reason, the possible signal transduction pathway that modulates Sema3A expression in PDLF and osteoblasts is also of interest. Data on the mechanoregulation of Sema3A are sparse. However, there is evidence that Sema3A could be a direct transcriptional target of the Osterix (SP7) transcription factor [15]. Furthermore, Osterix has been shown to be induced at the tension side in an animal model for orthodontic tooth movement [16].

Our own research on members of the Ephrin/Eph family and data on Sema3A functions for bone remodeling suggest that the role of neuronal guidance molecules in the regulation of orthodontic tooth movement might have been underestimated so far. Thus, the objectives of this project were (i.) to investigate if different mechanical forces modulate the expression of Sema3A and its receptors Neuropilin-1 and Plexin A1 in periodontal fibroblasts and osteoblasts of the alveolar bone, (ii.) to study the mechanical modulation of Sema3A, (iii.) to investigate the effects of Sema3A stimulation on alveolar bone osteoblasts differentiation, (iv.) to elucidate the potential molecular mechanisms underlying Sema3A-dependent osteoblast differentiation and finally, (v.) to test for a potential impact of Sema3A on alveolar bone osteoblast adhesion and motility.

In summary, this study was the first to show a differentially mechanically regulated modulation of Sema3A in periodontal fibroblasts. Alveolar bone osteoblasts responded to Sema3A stimulation with a Rac1 and β-catenin-dependent induction of differentiation markers. Together with our previous results on the Ephrin/Eph family, the current data suggest that neural guidance molecules are involved in the regulation of bone remodeling during orthodontic tooth movement. It is currently speculative whether such in vitro findings can be used clinically.

## 2. Results

### 2.1. Transcription of Semaphorin 3A and Its Receptors Is Differentially Regulated by Strain and Compressive Forces in Human Fibroblasts of the Periodontal Ligament

A decisive prerequisite indicating the participation of Semaphorin 3a in the control of bone remodeling during orthodontic tooth movement would be its modulation by mechanical forces in the cells of the periodontium (human primary fibroblasts of the periodontal ligament (hPDLF) and human primary osteoblasts of the alveolar bone (hOB). Thus, the mechanical regulation of SEMA3A expression and its receptors Neuropilin 1 (NRP1) and Plexin A1 (PLXNA1) in hPDLF and hOB was investigated. The respective cells, hPDLF and hOB, were obtained from six individual patients, i.e., evaluation was not performed on pairs of hPDLFs and hOBs from the same patient.


*Analysis of mRNA expression in hPDLF*


Three independent populations of hPDLF were examined by means of quantitative PCR after the application of mechanical strain (2%, 4, 24, 48, 72 and 96 h) or compression forces (30 g/cm^2^, 1, 6 and 10 h). In PDLF, significantly changed expressions of SEMA3A and its differential regulation by strain or compressive forces, respectively, were apparent. SEMA3A was significantly induced by strain after 4 and 72 h of strain and reached baseline levels after 96 h of strain, while compression forces led to a significant reduction of SEMA3A mRNA expression at all time points. PLXNA1 showed a significant induction only after 4 h of strain and was elevated after 24 h (n.s.) before, at 48 h to 96 h, mRNA expression was below baseline levels. PLXNA1 was not significantly altered after compression. NRP1 was slightly induced after 4 h of strain (n.s.) but was significantly reduced after 48 h of strain and remained below baseline levels after 72 h and 96 h of strain (both n.s.) and of compression forces. (Figure 1A–C).


*Analysis of protein expression in hPDLF*


Analyses of protein expression by Western blot confirmed the strain dependent induction of Sema3A and suggested differentially regulated receptor expressions: the induction of bot, Plexin A1 and Neuropilin 1 by strain and reduction by compression. Moreover, Sema3A protein expression was reduced by compression at the protein level (Figure 2A–C).

We observed incongruent expression between mRNAs and proteins for PlexinA1 and Neuropilin 1 during strain, as well as for PlexinA1 during compression. In addition, the mRNA and protein expressions of Sema3A showed a temporally different course.


*Analysis of mRNA expression in hOB*


Interestingly, no significant changes regarding the expression of SEMA3A and its receptors could be observed after the application of mechanical force in hOB (Figure 3A–D). In view of these data on mRNA expression modulation by mechanical forces of Sema3a and its receptors in hOB, we did not expect any significant changes in protein expression and therefore did not perform an analysis of protein expression for hOB.

### 2.2. Mechanical Strain Induced the Transcription Factor Osterix (SP7) in hPDLF

Based on existing evidence that Sema3A could be a direct transcriptional target of the Osterix (SP7) transcription factor [15] and data derived from an animal model of orthodontic tooth movement, wherein it was shown that Osterix can be induced on the tension side [16], we speculated that strain, Osterix and Sema3A expression might be associated in hPDLF. RT-qPCR analyses for Osterix in hPDLF subjected to mechanical strain (2.5%) or compressive forces (30 g/cm^2^) showed a significant induction of Osterix for up to 24 h in stretched cells, whereas compressed hPDLF showed significantly reduced Osterix expression (Figure 4). Although the mechanism that leads to its mechano-induction remains unclear, our results suggest that Osterix might be involved in mechano-induced Sema3A upregulation.

### 2.3. Exogenous Semaphorin 3A Stimulation Induced Osteogenic Marker Expression in hOB

To test the effect of Sema3A on the osteogenic differentiation of osteoblasts of the alveolar bone, three independent populations of hOB were stimulated with osteogenic medium with or without recombinant human Semaphorin 3A [10 ng/mL].

As shown in Figure 5A–D, exogenous Sema3A significantly stimulated the expression of the pivotal osteogenic transcription factor RUNX2 (Figure 5B). Other osteogenic marker genes (ALPL, alkaline phosphatase, Figure 5B; SPP1, osteopontin, Figure 5C; and BGLAP, osteocalcin, Figure 5D) showed a trend towards induction after stimulation with Sema3A beyond the effect of the osteogenic medium (mineralization medium: min. med. = DMEM, supplemented with 50 mg/mL ascorbic acid, 10 mM ß-glycerophosphate, 10-7 M dexamethasone). However, the effect of Sema3A on osteogenic gene expression was undetectable without osteogenic preconditioning (data not shown). Thus, mechano-dependent release of semaphorin 3A from PDLF might contribute to the osteogenic differentiation of osteoblasts of the alveolar bone only in an already osteogenic environment.

### 2.4. Semaphorin 3A-Dependent Osteogenic Marker Gene Induction in hOB Is Associated with Rac1 GTPase Activation and the Nuclear Translocation of β-Catenin

Data from mouse experiments suggest that Plexin-associated GTPases and the Wnt/β-catenin signaling pathway are involved in Sema3A-activated signal transduction in bone. Therefore, we tested if Sema3A stimulation leads to GTPase Rac1 activation and accumulation and the nuclear translocation of β-catenin in hOB. First, the involvement of the Rac1GTPase was checked with pull-down experiments. As shown in Figure 6A, stimulation of osteoblasts of the alveolar bone with exogenous Sema3A led to the activation of Rac1GTPase. Activation typically peaked 5–15 min after stimulation.

The observed Sema3A-dependent Rac1 activation in hOB prompted us to test for β-catenin contribution. RT q-PCR revealed a significant Sema3A contribution to β-catenin induction in hOB grown in mineralization medium (Figure 6B). For β-catenin-dependent transcriptional activation, the nuclear translocation of the protein is essential. To test for Sema3A-dependent nuclear translocation of β-catenin, immunofluorescence staining for β-catenin was performed in hOB stimulated with exogenous Sema3A [250 µg/mL]. In control cells, β-catenin was mainly found in the cytoplasm. Stimulation with Semaphorin 3A led to a perinuclear accumulation of β-catenin after 2 h. After 48 h, most of the β-catenin was translocated into the nucleus (Figure 6C). Based on these results, we speculate that the Sema3A-dependent induction of osteogenic marker gene expression in hOB occurs via a Rac1GTPase- and β-catenin-dependent signaling pathway.

### 2.5. Exogenous Semaphorin 3A Does Not Alter Focal Adhesion Contacts in hOB

Semaphorin3A acts as a repulsive signal for several types of developing neurons [17]. In several cell types, on binding to Neuropilin-1 and its co-receptor PlexinA1, Sema3A activates a signal transduction cascade that controls F-actin dynamics and thus cell adhesion and motility [18]. Altered matrix adhesion and thus, effects on cellular motility after Semaphorin 3A signaling are indicated by alterations in focal adhesion contacts. Recruiting of osteoblast precursors and osteoblasts to active remodeling sites during bone remodeling requires the increased motility of these cells. To test if alveolar bone osteoblast motility is affected by Semaphorin 3A, hOB was treated with recombinant Sema3A (100 ng/mL) for different time periods. Immunofluorescent staining for the focal adhesion protein vinculin was used to assess focal contacts. Staining against vinculin showed no evidence of altered focal contacts in the osteoblasts of the alveolar bone (Figure 7). From these data, it appears that Sema3A might not have a significant effect on osteoblast motility in the alveolar bone.

## 3. Discussion

The complete elucidation of the molecular biology of orthodontic tooth movement remains a goal to be achieved. Accumulating evidence suggests that neuronal guidance molecules might also be involved in the regulation of bone remodeling during orthodontic tooth movement.

### 3.1. What Is Known about the Role Neuronal Guidance Molecules in Bone Remodeling and Orthodontic Tooth Movement?

Initially, neuronal guidance molecules were identified to function in axon guidance and neuronal compartmentalizing [19], where they primarily control changes in cell motility and convey attractive and repulsive signals. Intensive research on these molecules has revealed numerous other functions to date. Neuronal guidance molecules are involved in angiogenesis [20], cardiogenesis [21], the regulation of immune cells [22], oncogenesis [23] and the regulation of bone homeostasis [24,25]. Functions in bone homeostasis have so far been described for neuronal guidance molecules from various families, including Ephrin-A2 [10], Ephrin-B2 [11], Sema3A [12], Slit 2 [26] and Netrin-1 [27].

Representatives from the family of the Ephrines and their Eph receptors, as well as members of the family of the Semaphorins with their receptors, the Neuropilins and Plexins, have so far been best characterized for functions to maintain bone homeostasis. We have previously shown a possible involvement of Ephrines and Eph receptors in bone remodeling during orthodontic tooth movement [13,14]. So far, however, there is no data on a possible role for the Semaphorins during tooth movement.

### 3.2. The Role of Sema3A in Bone Remodeling—Potential Implications for Orthodontic Tooth Movement—Hypotheses

Sema3A has been shown to be involved in embryonic bone development, and in vivo and in vitro data suggest a prominent involvement in adult bone remodeling [12,28,29]. Therefore, an alteration in Sema3A expression, stimulated mechanically by orthodontic forces, could play a role in orthodontic tooth movement by influencing bone remodeling. A strain-dependent modulation in hPDLF and gingival fibroblasts has already been shown, but the experimental strategy used did allow for a conclusive assessment [30]. Therefore, in the present study, we hypothesized that Sema3A, and/or its receptors Plexin A1 and Neuropilin 1, which are necessary for bone-specific signal transmission, are mechanically regulated in periodontal cells and that this regulation leads to the Sema3A-dependent induction of the osteoblasts of the alveolar bone.

Mechanical force plays an important role in the regulation of bone remodeling in intact bone and bone repair. Compression leads to cellular bent and/or compression, as opposed to a strain, which leads to tension, causing cells to be pulled apart. Different qualities of forces (strain vs. compression) have different consequences for bone cell mechanotransduction and thus, bone remodeling. During tooth movement, synchronized, albeit opposing, processes take place at the pressure and at the tension side. Such adaptive responses to the application of force include the reorganization of the intracellular and the extracellular matrix, the activation of various signal transduction events, alterations in gene expression and the release of molecular factors. This complex network of consecutive events leads to bone resorption on the pressure side and osteogenesis on the tension side, as teeth are moved within the alveolar bone [7,31]. To simulate mechanotransduction during orthodontic tooth movement wherein compression and strain cause different effects, it is advisable to investigate the effects of both strain (representing the tension side during orthodontic tooth movement) and compression (representing the pressure side). Therefore, in our current experiments, like others and ourselves in previous studies, we investigated the effects of both strain and compression as a model approaching orthodontic tooth movement in vivo.

During orthodontic tooth movement, stimulation of the osteoblast activity on the tension side would be expected, whereas on the compression side, bone resorption should occur. Due to its dual activity, both on osteoblasts and osteoclasts, we hypothesized a differential regulation of Sema3a by strain or compression forces.

The present study showed, by experiments on primary periodontal cells, that strain in the fibroblasts of the periodontal ligament significantly induced the expression of Sema3A and also affected its receptors Plexin A1 and Neuropilin 1. SEMA3A was significantly induced by strain after 4 and 72 h of strain and reached baseline levels after 96 h of strain, while compression forces led to a significant reduction of SEMA3A expression at all time points. PLXNA1 showed a significant induction only after 4 h of strain and was elevated after 24 h (n.s.) before, at 48 h to 96 h, mRNA expression was below base-line levels. PLXNA1 was not significantly altered after compression. NRP1 was slightly induced after 4 h of strain (n.s.) but significantly reduced after 48 h of strain and remained below baseline levels after 72 h and 96 h of strain (both n.s.) and of compression forces. Our exemplary analyses of protein expressions, however, partially showed a different expression or different kinetics of expression changes. We observed incongruent expression between mRNAs and proteins for PlexinA1 and Neuropilin 1 during strain, as well as for PlexinA1 during compression. In addition, the mRNA and protein expressions of Sema3A showed a temporally different course. At this point, we can only speculate about the causes. We would assume here that regulation at the mRNA level for the receptors occurred before the regulation of Semaphorin 3A mRNA. The mRNA regulation of PLXNA1 and NRP1 appears transient and seems to be tightly regulated (hence the marked to significant decrease after 48 h). Since at the protein level, the highest abundances of all proteins are reached only after 96 h, we would speculate that the processing and translation of the mRNAs for the receptors and Sema3A occur at different rates, and possibly the half-lives of the mRNAs of Sema3A and its receptors could also be different. In principle, however, we do not consider it impossible that the regulation of an mRNA expression temporally precedes the regulation of protein expression.

Compression forces, however, had no or opposite effects. Mechanically stimulated alterations of various Semaphorins have been described in vascular cells during development [32] and also in periodontal cells or osteoblasts [33]. While our results for Sema3A in periodontal fibroblasts are comparable with those obtained by Spencer et al. [30], interestingly, we did not identify significant modulation of Sema3A and its receptors in osteoblasts of the alveolar bone, which underlines the crucial role of periodontal fibroblasts in the regulation of orthodontic tooth movement.

Based on these results, the mechanical regulation of Semaphorin 3A in the periodontium appears to be limited to PDLF. However, regarding the pivotal importance of PDLF for the initiation and regulation of tooth movement, Sema3A modulation in PDLF might be associated with changes in bone remodeling in adjacent alveolar bone osteoblasts.

With regard to the regulatory pathways that integrate mechanical forces and Sema3a modulation in hPDLF, we have identified an association of the transcription factor Osterix (SP7) with strain and increased Sema3A expression in hPDLF. Osterix was identified previously as a zinc-finger-containing transcription factor, which plays an important role in osteoblast differentiation and bone formation. Nakashima et al., for instance, showed that Osterix knock-out mice showed no bone formation [34]. Microarray analyses on ATDC5 chondrocytes performed by Park et al. have identified Sema3A as a transcriptional target of Osterix [15]. Based on the association of Osterix and Sema3a expression in hPDLF, Osterix-dependent Sema3A activation seems plausible. However, previous data from mice deficient for odd-skipped related 2 transcription factor (Osr2), which show impaired bone development manifested by disrupted palatal growth and morphogenesis, showed Osr2 deficiency caused Sema3A induction; however, Sema3A induction in these Osr2-deficient animals was accompanied by simultaneous Osterix induction [35], which confirms an association between Osterix and Sema3A, which, however, might be regulated upstream by Osr2.

The data currently available on the mechano-dependent regulation of Semaphorins is still sparse, and the proposed mechanical, Osterix-dependent regulation of Sema3A signaling in periodontal cells certainly needs further proof.

The role of Sema3A in the regulation of bone remodeling is increasingly recognized [36] and also includes functions in the periodontium. Lin et al. found a reduced expression of Sema3A and Nrp1 in apical periodontitis, which suggests an involvement of Sema3A-dependent signaling in periapical bone resorption [37]. Using the physiological initiation of bone remodeling in the rat mandible as a model, Hassan et al. observed that a temporary loss of Sema3A expression was associated with the migration and activation of osteoclast precursor cells [38]. For the differentiation of odontoblasts from stem cells of the dental pulp, which are functionally similar to osteoblasts, differentiation triggered by Wnt/β-catenin signaling after Sema3A-dependent Nrp1 activation has been shown by Song and colleagues [39].

In order to test the specific effects of Sema3a on the osteoblasts of the alveolar ridge, we stimulated the osteoblasts of the alveolar ridge with recombinant human Sema3a. We found that the stimulation had an additive effect on the expression of osteogenic marker genes representing different stages of osteoblast differentiation; most importantly, we found a significant induction of RUNX2 expression after Sema3A stimulation. These findings are in line with previous research on osteoblasts of different origins and osteoblast precursor cells [12,36,40].

Interestingly, in our experiments, stimulation with Sema3a alone was not sufficient to induce osteogenic marker gene expression. Sema3 only had an effect on osteoblasts of the alveolar bone simultaneously stimulated with osteogenic medium (mineralization medium). This suggests that, for Sema3A function in bone remodeling during tooth movement, an osteogenic environment is necessary and that Sema3A may therefore not be able to initiate bone remodeling itself.

Data from murine models suggest that Plexin-associated GTPases and the WNT/β-catenin signaling pathway are involved in Sema3A signaling in bone [12]. In the canonical, β-catenin-dependent signaling pathway, the absence of the Wnt signal was shown to lead to a degradation of the β-catenin protein in the cytoplasm. This is ensured by the various proteins containing breakdown complex (APC). This constellation leads to phosphorylation and thus enables the ubiquitination of β-catenin, which is finally degraded in the proteasome. This state is reversed by the interaction of Wnt ligands (e.g., Wnt3A) with their heterodimeric receptor (Fzd/LRP). Subsequently, the accumulation and finally translocation of the hypophosphorylated β-catenin into the cell nucleus is initiated. In the nucleus, the expression of Wnt target genes such as RUNX2 is induced [41]. The activation of the PlexinA1 receptor by Sema3A contributes to the Wnt-induced accumulation and translocation of β-catenin via the PlexinA1-dependent activation of the Rac1GTPase [12,36].

Thus, we tested for GTPase and β-catenin contributions in Sema3A signaling in the osteoblasts of the alveolar bone. To this end, we first checked whether the Rac1GTPase is also involved in the early signal transduction after Neuropilin 1/Plexin A1 activation using pull-down experiments. Furthermore, β-catenin upregulation and nuclear translocation were assayed. We identified Rac1GTPase activation, increased β-catenin expression and its translocation to the nucleus which suggested that Sema3A downstream signaling in osteoblasts of the alveolar bone is dependent on the Wnt-pathway, which agrees with previous findings obtained from osteoblasts of other origins [12,40].

In the nervous system, many functions of neural guidance molecules, including Sema3A, are linked to changes in cellular adhesion and cellular motility. In several cell types, on binding to Neuropilin-1 and its co-receptor PlexinA1, Sema3A activates a signal transduction cascade that controls F-actin dynamics and thus, cell adhesion and motility [18]. Bone remodeling requires osteogenic cells to reach the sites of remodeling, and it is recognized today that osteoblasts have migratory potential [42]. We therefore wanted to check whether Sema3a could possibly play a role in the adhesion and thus, the motility of osteoblasts of the alveolar ridge. However, we found that, in osteoblasts of the alveolar bone, Sema3A stimulation did not lead to perturbed focal adhesion contacts, suggesting a minor to negligible role of Sema3A in osteoblast motility in bone remodeling during orthodontic tooth movement in the alveolar bone.

During this study, we used different concentrations of recombinant Semaphorin 3a for different experimental approaches. The individual concentrations were titrated in advance for each experiment. We do not think that this has resulted in a major problem for the interpretation of our data. From our point of view, it should be considered that the respective incubation times are very different (between minutes and days), so that a comparison of the integrals (concentration × time) makes the differences appear minimal.

### 3.3. Conclusions

Our results for osteoblasts of the alveolar bone correspond to the current state of research, which increasingly accepts an important function of Sema3A-dependent signaling for dental hard tissue formation and bone remodeling in the periodontium and in vitro [43,44]. From previous studies, it remained unclear what effects mechanical forces might have on Sema3A signaling. Our study is the first to show a connection between mechanical force and the modulation of Sema3A and its receptors in periodontal fibroblasts. Mechano-regulation of Sema3A signaling might not necessarily be limited to bone remodeling during orthodontic tooth movement, since essential remodeling processes that serve to maintain the bone mass are mechanically induced skeleton-wide. In the case of bone remodeling induced by orthodontic forces, however, based on the results presented here, periodontal fibroblasts, as recipients of the mechanical forces, seem to be of primary importance. Based on the differential mechanical regulation of Sema3A, we suggest a possible involvement of Sema3A signaling during the initiation of bone remodeling after the application of orthodontic forces. We could confirm Sema3A functions observed in osteoblasts of other origins in osteoblasts of the alveolar bone but in a weakened form and only under additional osteogenic stimulation, which possibly limits the importance of Sema3A alone for remodeling the alveolar bone. However, together with our previous studies on the Ephrin/Eph family, members of families of neuronal guidance molecules that are still expressed in adult dental tissue could have functions in maintaining bone homeostasis, regulating mechano-induced bone remodeling and possibly controlling the regenerative potential of periodontal tissues.

However, at this time we see no clear perspectives for clinical application of Sema3A. Animal models are urgently required to confirm in vitro findings and to determine the potential undesirable effects of Sema3A application. In addition, the use of Sema3A in enhancing the development and in vitro differentiation of dental hard tissue merits observation.

## 4. Materials and Methods

### 4.1. Primary Cell Cultures

Primary PDL fibroblasts were obtained from juvenile patients (12–20 years) following premolar extraction indicated during orthodontic treatment. Alveolar bone tissue was obtained from patients following osteotomy of third molars. The Ethics Committee (Medical Faculty, University of Heidelberg; Vote S-071/2016) has approved the harvest of the tissues. Informed consent was obtained from the patients following an explanation of the study.

Fragments of PDL tissue or small bone tissue fragments were washed to remove excess blood with an equal volume of phosphate-buffered saline. The tissue was then transferred to a Petri dish, where it was minced into smaller fragments. These tissue fragments were evenly distributed over the surface of tissue culture dishes and cultured in DME medium (Thermo Fisher Scientific, Karlsruhe, Germany) supplemented with 10% fetal calf serum (FCS), 2 mM L-glutamine, antibiotics and antimycotics. Primary PDL cells and primary osteoblasts were used between passages 3 to 7 for the experiments.

For Semaphorin 3A stimulation experiments, cells were starved for 24 h in culture medium and supplemented with 2% FCS (starvation medium). Stimulation with recombinant human Sema3A (R&D Systems, Wiesbaden, Germany) was done in starvation medium at concentrations and for durations as indicated in the individual experiments.

Osteogenic induction was performed by culturing the cells in mineralization medium [DMEM, supplemented with 50 mg/mL ascorbic acid, 10 mM ß-glycerophosphate (Calbiochem, Germany), 10-7 M dexamethasone (Calbiochem, Germany)].

### 4.2. Application of Static Compression and Static Strain

Static compression to PDLF was simulated via centrifugation. Confluent cultures of PDLF were centrifuged in cell culture dishes (60 mm) at 127 g in a swing out rotor (Beckman GH3.7) for 1, 4 or 6 h, respectively. The compressive force applied to the cells was 30.3 g/cm^2^, simulating clinical orthodontic forces (Davidovitch, 1991). The applied force was calculated according to the following formula: *P* = (m × r × rpm^2^ × π^2^)/A * 9.8 * 900), *P* = pressure [kg/cm^2^], m = mass (of medium) [g], r = radius [m], A = surface area of culture dish [cm^2^]. The application of compressive forces on PDLF has been established previously (Redlich et al., 2004). Centrifugal forces to simulate compression in PDLF have been successfully used by others (Kook et al., 2009) and our own laboratory. Strain application was carried out according to the method described by Hasegawa et al. [45]. Briefly, 5 × 103/cm^2^ PDL or osteoblast cells were seeded on Lumox flexible bottom dishes (Greiner Bio-One, Frickenhausen, Germany), coated with 20 μg/mL collagen type-I Merck Millipore, Darmstadt, Germany) and 10 μg/mL fibronectin (Biomol, Hamburg, Germany) and grown until 80% confluence. The bottom of each dish was strained by induction of a continuous average strain of 2.5% for the periods indicated for the individual experiments (see results section for details). Unstrained cells served as controls.

### 4.3. Quantitative RT-PCR Analysis

Total RNA was isolated from cells using the RNeasy-Kit (Qiagen; Hilden, Germany). RNA integrity was monitored by capillary electrophoresis (Experion System, Bio-Rad, Munich, Germany). Total RNA was subjected to reverse transcription using poly-dT-Primers. Single-stranded cDNA was used for qPCR analyses. Quantitative PCR was performed using SYBR Green chemistry on an iCycler Instrument (Bio-Rad, Munich, Germany). To ensure equal amplification efficiencies, we used predesigned RT^2^ qPCR Primer Assays (Qiagen, Hilden, Germany). See Table 1 below for assay IDs. The relative gene expression was determined using the delta-delta CT method [46].

### 4.4. Western Blotting and Immunofluorescence Staining

Protein lysates were obtained by lysis with ice cold RIPA buffer supplemented with protease inhibitor cocktail (Roche, Mannheim, Germany). Moreover, 25 µg of protein were separated on 4–12% NuPAGE Gels (Invitrogen, Karlsruhe, Germany. The separated proteins were transferred onto PVDF membranes (Invitrogen, Karlsruhe, Germany). Membranes were probed with antibodies against Semaphorin 3A (Clone #215803, 1 µg/mL, R&D Systems, Wiesbaden, Germany), Plexin A1 (Clone #659807, 1 µg/mL, R&D Systems, Wiesbaden, Germany) and Neuropilin 1 (D62C6, 1:2000; Cell Signaling/New England Biolabs, Frankfurt, Germany). The Western Breeze Chromogenic Immunodetection Systems (Life Technologies, Darmstadt, Germany) against mouse or rabbit primary antibodies was used for visualization. Blots were scanned and densitometric analyses performed using ImageJ [47].

For immunostaining, cells were grown on Lumox dishes. After Semaphorin 3A stimulation, cells were washed with ice cold PBS, fixed for 5 min with methanol/acetone (1:1, −20 °C) and air-dried. Lumox membranes were cut into parts and staining was performed in multiwall plates. Incubation with the antibody against β-catenin (Clone #196618, 8µg/mL, R&D Systems, Wiesbaden, Germany) or vinculin (Clone# 42H89L44, 1µg/mL, ThermoFisher Scientific, Karlsruhe, Germany) was performed overnight at 4 °C. After washing, Lumox membranes were incubated with Dylight 488 fluorochrome-conjugated antibodies (Jackson ImmunoResearch, Dianova, Hamburg, Germany) for 1 h at room temperature. DAPI was used as a nuclear counterstain. For negative controls, the primary antibodies were omitted. Microphotographs were taken using a Leica DMRE microscope equipped with a digital camera (DFC300 FX, Leica, Bensheim, Germany). Image-acquisition and processing was done using the Leica Application Suite software (Leica, Bensheim, Germany).

### 4.5. Rac1GTPase Activity Assay

Rac1 activity (GTP-Rac1) was evaluated using a commercially available Rac1 activation assay based on GST (glutathione-S-transferase)–CRIP (Cdc42/Rac Interactive Binding) fusion proteins, which specifically bind the GTP-bound (=active) form of Rac1GTPase (Merck-Millipore, Darmstadt, Germany), according to the manufacturer’s instructions.

### 4.6. Statistical Analysis

Results are presented as mean ± standard deviation (SD). Differences between treatments were compared using a Kruskal–Wallis one-way analysis of variance on ranks (ANOVA), followed by Dunn’s post-hoc test. All statistics were performed using SigmaPlot software version 14 (Systat Software GmbH, Erkrath, Germany). Results were considered significant with a *p* value < 0.05.

## Figures and Tables

**Figure 1 ijms-22-08297-f001:**
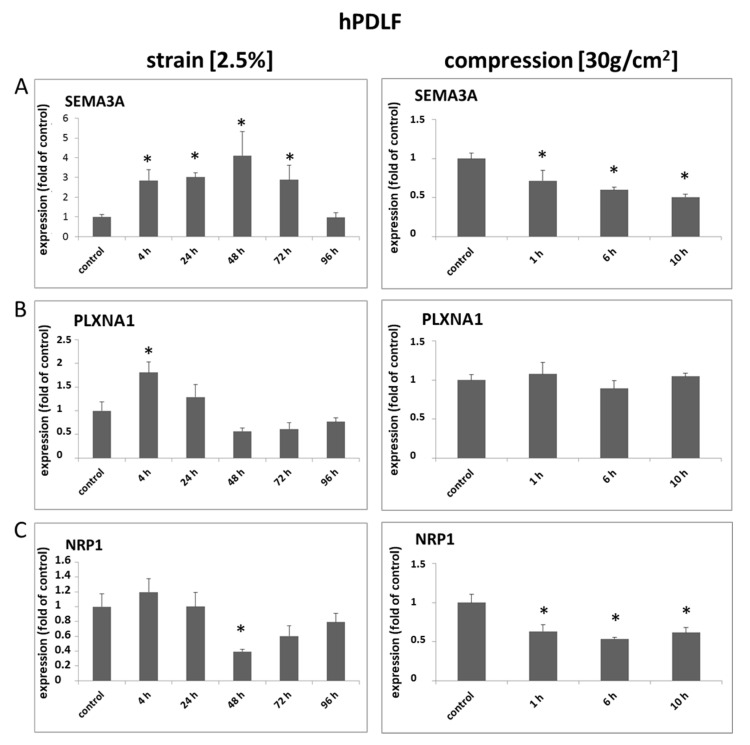
The mRNA expression of SEMA3A, PLXNA1 and NRP1 is differentially regulated from mechanical compression or strain. Three independent primary human PDLF (hPDLF) populations were either stretched (2.5%) or compressed (30 g/cm^2^) for the indicated periods of time. Stretching and compression had opposite effects on the expressions of SEMA3A (**A**), PLXNA1 (**B**) and NRP1 (**C**). Experiments were carried out in triplicates. Untreated cells served as controls. Quantitative RT-PCR experiments were carried out in triplicates. Cumulative data of the three cell populations data are given as mean ± SD. * = *p* < 0.05 vs. control (Kruskal–Wallis one-way analysis of variance on ranks (ANOVA), followed by Dunn’s post-hoc test).

**Figure 2 ijms-22-08297-f002:**
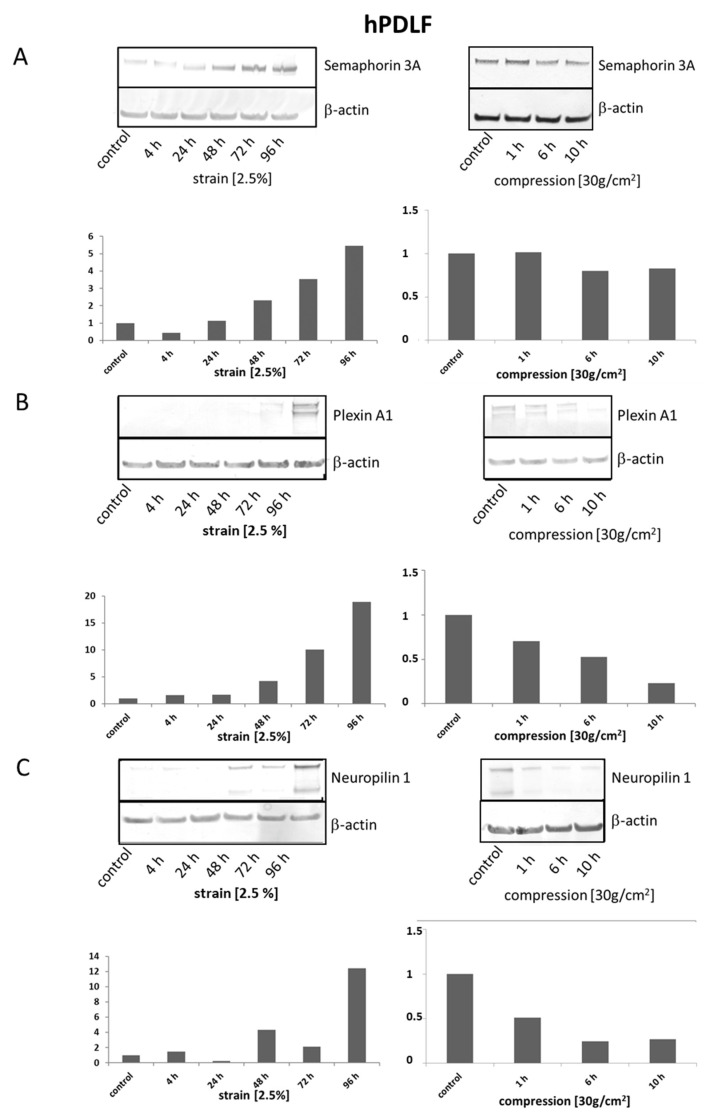
The protein expression of Sema3A, Plexin A1 and Neuropilin 1 is differentially regulated depending on mechanical compression or stretching. Primary human PDLF populations were either stretched (2.5%) or compressed (30 g/cm^2^) for the indicated periods of time. At the protein level, stretching and compression also had opposing effects on the expressions of Sema3A (**A**), Plexin A1 (**B**) and Neuropilin 1 (**C**) and thus support the results of the mRNA expression analyses. Representative Western blots are shown. Densitometric analyses, normalized to the expression of β-actin, which was detected on the same membrane, are added to illustrate protein expression changes.

**Figure 3 ijms-22-08297-f003:**
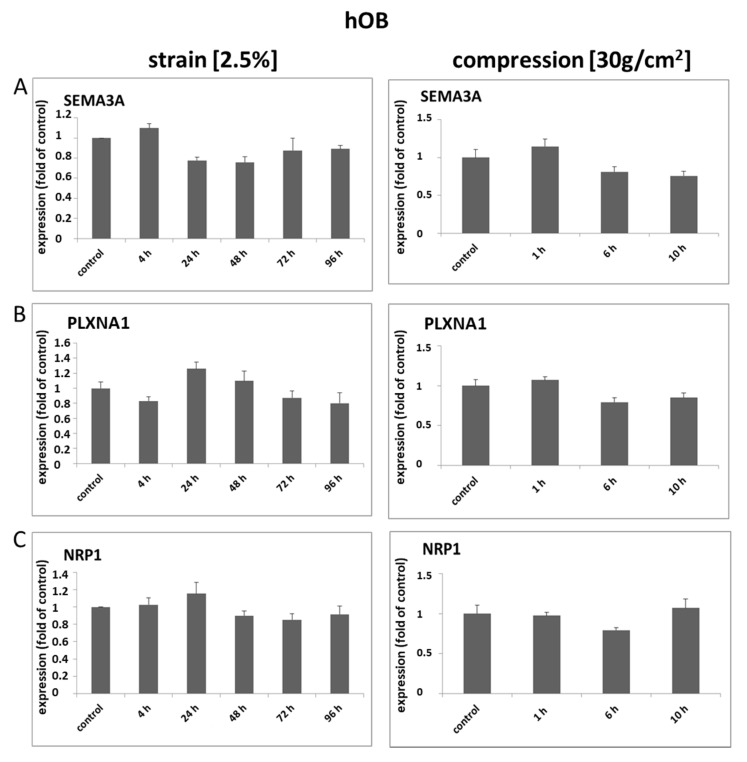
SEMA3A, PLXNA1 and NRP1 show no significant mechanically induced expression changes in osteoblasts of the alveolar bone. Three independent populations of primary human osteoblasts of alveolar bones were either strained (2.5%) or compressed (30 g/cm^2^) for the indicated periods of time. Neither strain nor compression had significant effects on the expressions of SEMA3A (**A**), PLXNA1 (**B**) and NRP1 (**C**). Untreated cells served as controls. Quantitative RT-PCR experiments were carried out in triplicates. Cumulative data of the three cell populations data are given as mean ± SD.

**Figure 4 ijms-22-08297-f004:**
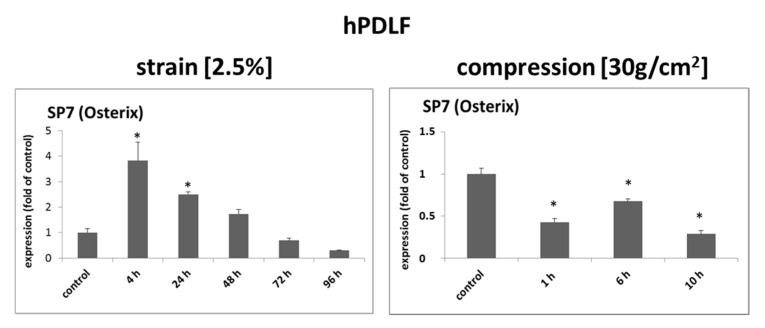
Osterix (SP7) is induced by mechanical strain in human primary periodontal fibroblasts. Three independent primary human PDLF (hPDLF) populations were either stretched (2.5%) or compressed (30 g/cm^2^) for the indicated periods of time. Stretching and compression had opposite effects on the expressions of Osterix (SP7). Experiments were carried out in triplicate. Untreated cells served as controls. Quantitative RT-PCR experiments were carried out in triplicate. Cumulative data of the three cell populations data are given as mean ± SD. * = *p* < 0.05 vs. control (Kruskal–Wallis one-way analysis of variance on ranks (ANOVA), followed by Dunn’s post-hoc test).

**Figure 5 ijms-22-08297-f005:**
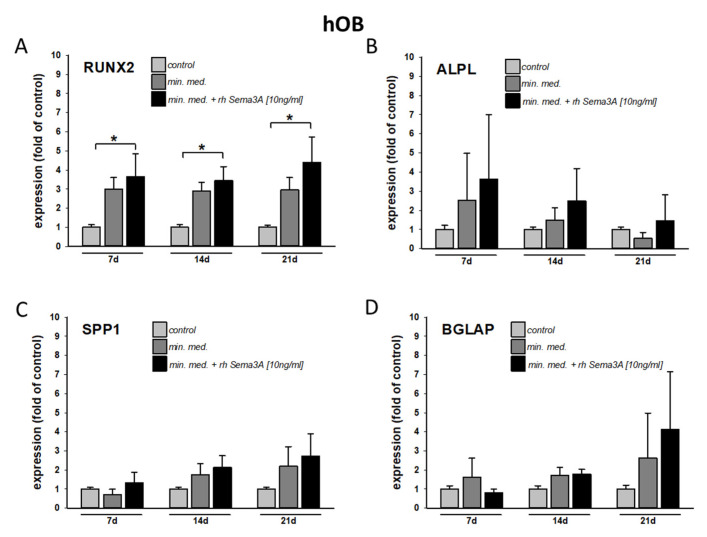
Exogenous Semaphorin 3A stimulation contributes to the osteogenic differentiation of osteoblasts of the alveolar bone (hOB) in an already osteogenic environment. Three independent populations of primary human osteoblasts of the alveolar bone were cultivated with osteogenic differentiation medium with and without the addition of recombinant Semaphorin 3A [10 ng/mL] for 7–21 days. After 14 and 21 days, Semaphorin 3A increased the expression of osteogenic marker genes: RUNX2 (**A**), ALPL (**B**), SPP1 (**C**) and BGLAP (**D**). Experiments were carried out in triplicate. Untreated cells served as controls. Quantitative RT-PCR experiments were carried out in triplicate. Cumulative data of the three cell populations data are given as mean ± SD. * = *p* < 0.05 vs. control (Kruskal–Wallis one-way analysis of variance on ranks (ANOVA), followed by Dunn’s post-hoc test).

**Figure 6 ijms-22-08297-f006:**
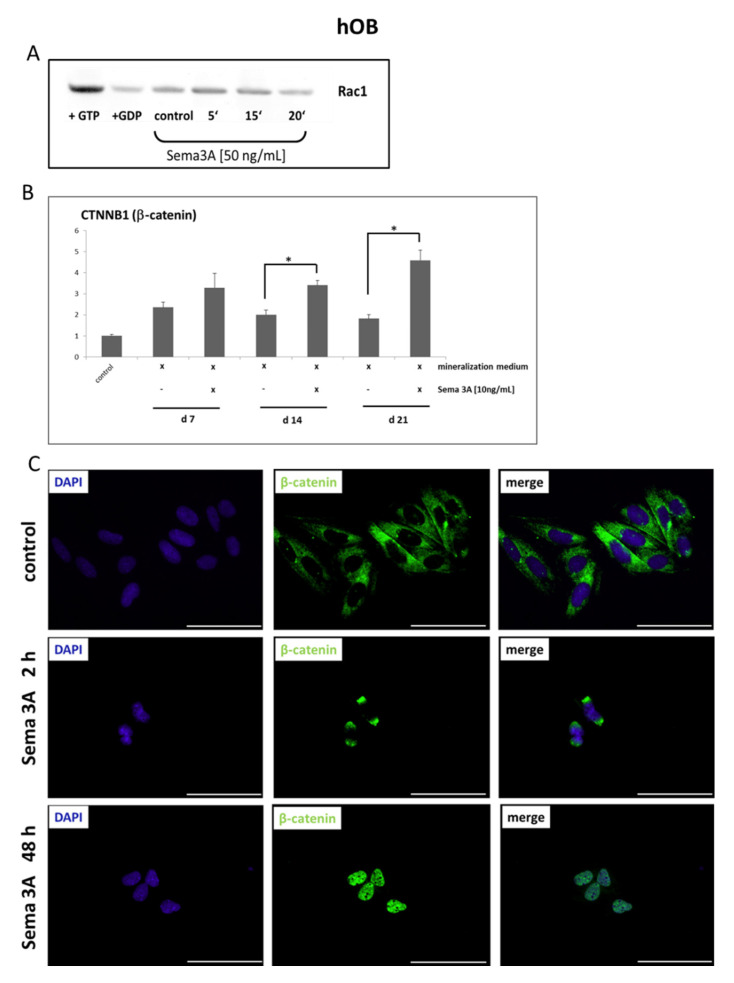
Semaphorin 3A stimulation of human primary osteoblasts of the alveolar bone (hOB) is associated with Rac1GTPase activation, as well as transcriptional induction and the nuclear translocation of β-catenin: (**A**) Osteoblasts of the alveolar bone were stimulated with recombinant Sema 3A [50 ng/mL] for the periods indicated. Pull down was carried out with GST (glutathione-S-transferase)—CRIP (Cdc42/Rac interactive binding) fusion proteins. A representative Western blot shows that exogenous Sema3A activated Rac1GTPase after 5–15 min. before returning to baseline levels after 20–30 min. (**B**) Populations of primary human alveolar osteoblasts were cultivated with osteogenic differentiation medium with and without the addition of recombinant Sema 3A [10 ng/mL] for 7–21 days. Semaphorin 3A significantly increased the expression of β-catenin (CTNNB1), suggesting a possible contribution of Semaphorin 3A to the osteogenic differentiation of human alveolar osteoblasts. Untreated cells served as controls. Quantitative RT-PCR experiments were carried out in triplicate. Cumulative data of the three cell populations data are given as mean ± SD. * = *p* < 0.05 vs. control (Kruskal–Wallis one-way analysis of variance on ranks (ANOVA), followed by Dunn’s post-hoc test). (**C**) Primary human alveolar osteoblasts were stimulated with recombinant Semaphorin 3a [250 ng/mL] for up to 48 h. In untreated cells (control, upper panel), β -catenin was mainly located in the cytoplasm. The stimulation with Semaphorin 3A led to a perinuclear accumulation of β -catenin after 2 h (Sema3A 2 h, middle panel). After 48 h, most of the β -catenin was translocated into the nucleus (Sema3A 4 h, lower panel). Representative immunofluorescence staining for β-catenin (green) on hOB. DAPI (blue) = 4’, 6-diamidino-2-phenylindole. Bar = 20 µm.

**Figure 7 ijms-22-08297-f007:**
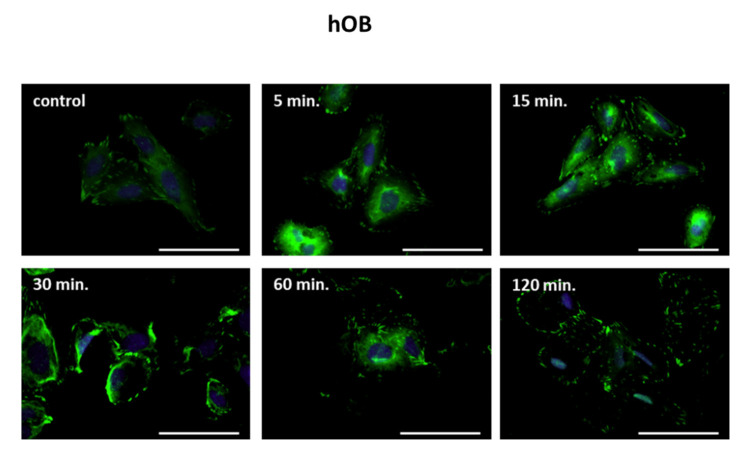
Exogenous Sema3A does not lead to alterations of the focal contacts in osteoblasts of the alveolar bone. Human primary osteoblasts of the alveolar bone were stimulated with recombinant Sema 3A [100 ng/mL] for the periods indicated. The detection of vinculin by immunofluorescence did not reveal any Sema3A-dependent dissolution of focal contacts. Scale = 20 µm.

**Table 1 ijms-22-08297-t001:** Quantitative PCR Primer Assays.

Target	Assay ID (GenGlobe ID)
human SEMA34	PPH05966F
human PLXNA1	PPH20083A
human NRP1	PPH01152A
human CTNNB1	PPH00643F
human SP7 (osterix)	PPH00705A
human RUNX2	PPH01897C
human ALPL	PPH58134F
human BGLAP (osteocalcin)	PPH01898A
human SPP1 (osteopontin	PPH00582E

## Data Availability

The data presented in this study are available on request from the corresponding author.

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
