# Peer review of "A Potential Role of Semaphorin 3A during Orthodontic Tooth Movement"

_ijms, 2021, doi:10.3390/ijms22158297_

Round 1

Reviewer 1 Report

  1. Fig.2 B: A clear PlexinA1 and NRP1 expression/signal is visible in control samples during compression, but was totally missing in the strain western blot. Why?
  2. Why there was a significant reduction in Sema3A receptors PLXNA1 and NRP1 during 48h of strain, which was actually opposite to the expression of Sema3A mRNA at the same time point?

Further, during strain, expression of Sema4A, PlexinA1, and NRP1 follow the same expression pattern (starting to increase at 48h and gradual increase up to 96h) complete contradiction to qRT-PCR data. Authors need to explain and discuss the reasons.

  1. Why did the authors change rhSema3A concentration in each experiment? Four different increasing concentrations (10, 50, 100, and 250 ng/ml) of rhSema3A were used on hOBs in different experiments as depicted in Figs. 5, 6, and 7? Why there is no uniformity in the rhSema3A concentration used? The results cannot be compared and concluded with this kind of concentration variation in different experiments.
  2. With so much SD in Fig 5, it is difficult to predict an increase in ALPL, BGLAP expression (clearly not significant). With no significance in p-value, how can the authors claim that there is an upregulation?
  3. Osterix has been defined as SP7 multiple times in the manuscript which has to be done in principle only once at the first time.
  4. What is the effect of Sema3A inhibitors on the osteogenic potential of hOBs? Does this result in the baseline level of osteogenic markers expression?

This experiment is necessary in order to confirm the role of PDLF-derived Sema3A on osteogenic differentiation of osteoblasts of the alveolar bone.

Author Response

Again, we thank the reviewers for the close reading of our manuscript for the interest in our work and for the valuable suggestions for improvement! At the same time, we would like to thank Reviewer 2 for the favourable review of our manuscript after the first revision!

We regret that the changes we have made so far and our point-to-point responses have not succeeded in convincing Reviewer 1 of the quality of our work.  

This is all the more regrettable since much of the present criticism is similar to that raised in the first review.

We therefore try to address the criticism again, but we also see limitations with regard to explainability and additional experimental effort, for which we would like to ask for your understanding already now.

Reviewer 1

  1. Fig.2 B: A clear PlexinA1 and NRP1 expression/signal is visible in control samples during compression, but was totally missing in the strain western blot. Why?

Our response:

We have, in response to criticism expressed about the original version of the manuscript, changed the exemplary figures for the analysis of protein expression. We have used available sample material as far as possible. In the case of the western blot for PlexinA1, to clarify the regulation beyond doubt, we chose a PDL population in which the baseline expression of the PlexinA1 receptor is below the detection limit of the Western blot set up we used. We further emphasize that the examples shown for strain and compression are illustrative and do not show identical populations in this case.

We understand the Reviewer’s concern, but we would like to point out again that, usually, when working with primary cell populations, heterogeneities must be assumed, which others and we usually try to compensate for by using multiple populations. We would like to emphasize that we consider these and the previously depicted western blots to be absolutely representative and are convinced of the regulation illustrated by the western blots.

No changes were made to the manuscript.

  1. Why there was a significant reduction in Sema3A receptors PLXNA1 and NRP1 during 48h of strain, which was actually opposite to the expression of Sema3A mRNA at the same time point? Further, during strain, expression of Sema4A, PlexinA1, and NRP1 follow the same expression pattern (starting to increase at 48h and gradual increase up to 96h) complete contradiction to qRT-PCR data. Authors need to explain and discuss the reasons.

Our response:

We would like to refer you to our reply to the original version:

We can only speculate on these results. We would assume here that regulation at the mRNA level for the receptors occurred before regulation of semaphorin 3A mRNA. The mRNA regulation of PLXNA1 and NRP1 appears transient and seems to be tightly regulated (hence the marked to significant decrease after 48 h). Since at the protein level the highest abundances of all proteins are reached only after 96 h, we would speculate that the processing and translation of the mRNAs for the receptors and Sema3A occur at different rates and possibly the half-lives of the mRNAs of Sema3A and its receptors could also be different.

In principle, however, we do not consider it impossible that the regulation of an mRNA expression temporally precedes the regulation of protein expression.

Therefore, we see no problem in accounting for the kinetics of mRNA and protein expression.

  1. Why did the authors change rhSema3A concentration in each experiment? Four different increasing concentrations (10, 50, 100, and 250 ng/ml) of rhSema3A were used on hOBs in different experiments as depicted in Figs. 5, 6, and 7? Why there is no uniformity in the rhSema3A concentration used? The results cannot be compared and concluded with this kind of concentration variation in different experiments.

Our response:

Again, we would like to refer you to our reply to the original version:

The individual concentrations were titrated in advance for each experiment. From our point of view, it should also be considered that the respective incubation times are very different (between minutes and days), so that a comparison of the integrals already makes the differences appear minimal.

No changes were made to the manuscript.

  1. With so much SD in Fig 5, it is difficult to predict an increase in ALPL, BGLAP expression (clearly not significant). With no significance in p-value, how can the authors claim that there is an upregulation?

Our response:

We follow the argumentation of the reviewer and have tried to express ourselves more clearly and have specified the description of our results by adapting the respective results section (changes are also highlighted in Yellow in the manuscript):

As shown in Fig. 5 A-D exogenous Sema3A significantly stimulated the expression of the pivotal osteogenic transcription factor RUNX2 (Fig. 5B). Other osteogenic marker genes (ALPL, alkaline phosphatase, Fig. 5B; SPP1, osteopontin, Fig. 5C; and BGLAP, osteocalcin, Fig. 5D) showed a trend towards induction after stimulation with Sema3A beyond the effect of osteogenic medium (mineralization medium: min. med. = DMEM, supplemented with 50 mg/ml ascorbic acid, 10mM ß-glycerophosphate, 10-7 M dexamethasone). However, the effect of Sema3a on osteogenic gene expression was undetectable without osteogenic preconditioning (data not shown). Thus, mechano-dependent release of semaphorin 3A from PDLF might contribute to osteogenic differentiation of osteoblasts of the alveolar bone only in an already osteogenic environment.

  1. Osterix has been defined as SP7 multiple times in the manuscript which has to be done in principle only once at the first time.

Our response:

We have chosen to use both terms throughout based on our review of numerous studies on Osterix. It is our understanding that while SP7 is the correct term, Osterix is still very widely used and is the most common term for most researchers in the field. However, in order to comply with nomenclature regulations, we have decided to use both designations in parallel.

No changes were made to the manuscript.

  1. What is the effect of Sema3A inhibitors on the osteogenic potential of hOBs? Does this result in the baseline level of osteogenic markers expression?

This experiment is necessary in order to confirm the role of PDLF-derived Sema3A on osteogenic differentiation of osteoblasts of the alveolar bone.

Our response:

We agree with the reviewer that blockade of strain-induced Sema3A expression has putative consequences for the proliferation, differentiation, and survival of osteoblasts and osteocytes.  However, However, both has been clearly demonstrated in the outstanding work of Hayashi and colleagues (2012, 2019) in various model systems.

We therefore see no need to explicitly test this again in our osteoblasts. We would like to emphasize again that the decisive and new result of our work is the mechanically regulated expression of the Sema3A signal transduction pathway in fibroblasts of the periodontal ligament (PDL-F) network and that the analyses we presented in osteoblasts are largely confirmatory in nature and are somewhat in the background compared to the results obtained in PDL Fibroblasts and should therefore not be further expanded.

We appreciate the reviewer's criticism and find it regrettable that he did not find our responses to the original version convincing. At the same time, however, we must state that we feel we have responded appropriately to the criticism on the original version of the manuscript.

We would be deeply sorry if the reviewer still does not find it possible to accept our answers, but we must admit that we do not have any better answers than the ones we have already given.

Reviewer 2 Report

The text has been revised according to my requests with significant overall  improvement so that the present version of the manuscript can be considered for publication. I have no further queries.

Author Response

Again, we thank the reviewers for the close reading of our manuscript for the interest in our work and for the valuable suggestions for improvement! At the same time, we would like to thank Reviewer 2 for the favourable review of our manuscript after the first revision!

Round 2

Reviewer 1 Report

These are the fundamental questions that we need to address to the scientific community to showcase that the results we show through our publications are in line with the figures that we present. In results, the authors generalized and claim that there is an increase/decrease which actually not seen/convincing with the figures they show. They should first stop generalizing their results statements and at least explain their observations (even minor differences) in detail (like at 48 hours we see some decline/increase which actually contradicting the observed result etc) and try to convince their arguments in the discussion. The authors should clearly write what they see in their results but not what they assume.

The response to point of increasing concentration of the rhSEMA3A used should be clearly written in discussion with the explanations they had given.

Author Response

Reply to Reveiwer 1

Comments and Suggestions for Authors

These are the fundamental questions that we need to address to the scientific community to showcase that the results we show through our publications are in line with the figures that we present. In results, the authors generalized and claim that there is an increase/decrease which actually not seen/convincing with the figures they show. They should first stop generalizing their results statements and at least explain their observations (even minor differences) in detail (like at 48 hours we see some decline/increase which actually contradicting the observed result etc) and try to convince their arguments in the discussion. The authors should clearly write what they see in their results but not what they assume.
The response to point of increasing concentration of the rhSEMA3A used should be clearly written in discussion with the explanations they had given.

Thank you for giving us another opportunity to improve our manuscript according to the reviewer's suggestions.

We have now tried to describe the results in more detail and have extended the discussion of the results and softened our conclusions on the role of Sema3A. We have also discussed the different concentrations of recombinant Sema3A used. We now hope to have met the requirements.

The following changes have been made to the text (these are also highlighted in yellow in the manuscript)

Results section (starting from line 124)

In PDLF, significantly changed expressions of SEMA3A and its differential regulation by strain or compressive forces, respectively, were apparent. SEMA3A was significantly induced by strain after 4 and 72h of strain and reached baseline levels after 96h of strain, while compression forces led to a significant reduction of SEMA3A mRNA expression at all time points. PLXNA1 showed a significant induction only after 4 h of strain and was elevated after 24h (n.s.) before at 48h to 96h mRNA expression was below baseline levels. PLXNA1 was not significantly altered after compression. NRP1, was slightly induced after 4h of strain (n.s.) but significantly reduced after 48h of strain remained below baseline levels after 72h and 96h of strain (both n.s.) and by compression forces. (Fig. 1 A-C).

Analysis of protein expression in hPDLF

Analyses of the protein expression by western blot confirmed the strain dependent induction of Sema3A and suggested differentially regulated receptor expressions: induction of both, Plexin A1 and Neuropilin 1, by strain and reduction by compression. Also, Sema3A protein expression was reduced by compression at the protein level (Fig. 2 A-C).

We observed incongruent expression between mRNAs and proteins for PlexinA1 and Neuropilin 1 during strain as well as for PlexinA1 during compression. In addition, the mRNA and protein expressions of Sema3A showed a temporally different course.

Results section (starting from line 209)

As shown in Fig. 5 A-D exogenous Sema3A significantly stimulated the expression of the pivotal osteogenic transcription factor RUNX2 (Fig. 5B). Other osteogenic marker genes (ALPL, alkaline phosphatase, Fig. 5B; SPP1, osteopontin, Fig. 5C; and BGLAP, osteocalcin, Fig. 5D) showed a trend towards induction after stimulation with Sema3A beyond the effect of osteogenic medium (mineralization medium: min. med. = DMEM, supplemented with 50 mg/ml ascorbic acid, 10mM ß-glycerophosphate, 10-7 M dexamethasone). How-ever, the effect of Sema3a on osteogenic gene expression was undetectable without osteo-genic preconditioning (data not shown). Thus, mechano-dependent release of semaphorin 3A from PDLF might contribute to osteogenic differentiation of osteoblasts of the alveolar bone only in an already osteogenic environment.

Discussion section (starting from line 356)

The present study showed, by experiments on primary periodontal cells, that strain in fibroblasts of the periodontal ligament significantly induced the expression of Sema3A and also affected its receptors Plexin A1 and Neuropilin 1. SEMA3A was significantly induced by strain after 4 and 72h of strain and reached baseline levels after 96h of strain, while compression forces led to a significant reduction of SEMA3A expression at all time points. PLXNA1 showed a significant induction only after 4 h of strain and was elevated after 24h (n.s.) before at 48h to 96h mRNA expression was below base-line levels. PLXNA1 was not significantly altered after compression. NRP1, was slightly induced after 4h of strain (n.s.) but significantly reduced after 48h of strain remained be-low baseline levels after 72h and 96h of strain (both n.s.) and by compression forces. Our exemplary analyzes of the protein expressions, however, partially showed a different expression or different kinetics of the expression changes. We observed incongruent expression between mRNAs and proteins for PlexinA1 and Neuropilin 1 during strain as well as for PlexinA1 during compression. In addition, the mRNA and protein expressions of Sema3A showed a temporally different course. At this point we can only speculate about the causes. We would assume here that regulation at the mRNA level for the receptors occurred before regulation of Semaphorin 3A mRNA. The mRNA regulation of PLXNA1 and NRP1 appears transient and seems to be tightly regulated (hence the marked to significant decrease after 48 h). Since at the protein level the highest abundances of all proteins are reached only after 96 h, we would speculate that the processing and translation of the mRNAs for the receptors and Sema3A occur at different rates and possibly the half-lives of the mRNAs of Sema3A and its receptors could also be different. In principle, however, we do not consider it impossible that the regulation of an mRNA expression temporally precedes the regulation of protein expression.

Compression forces, however, had no or opposite effects. Mechanically stimulated alterations of various Semaphorins have been described in vascular cells during development [32] and also in periodontal cells or osteoblasts [33]. While our results for Sema3A in periodontal fibroblasts are comparable with those obtained by Spencer et al. [30]. Interestingly we did not identify significant modulation of Sema3A and its receptors in osteoblasts of the alveolar bone which underlines the crucial role of periodontal fibroblasts in the regulation of orthodontic tooth movement.

Based on these results, mechanical regulation of Semaphorin 3A in the periodontium appears to be limited to PDLF. However, regarding the pivotal importance of PDLF for the initiation and regulation of tooth movement, Sema3A modulation in PDLF might be associated with changes in bone remodeling in adjacent alveolar bone osteoblasts.

Conclusion section (starting from line 488)

We could confirm Sema3A functions observed in osteoblasts of other origins in osteoblasts of the alveolar bone but in a weakened form and only under additional osteogenic stimulation, which possibly limits the importance of Sema3A alone for remodeling of the alveolar bone.

This manuscript is a resubmission of an earlier submission. The following is a list of the peer review reports and author responses from that submission.

Round 1

Reviewer 1 Report

The current work addresses a topic of interest for the Journal`s readers, since it presents an experimental research concerning the biology of orthodontic tooth movement by assessing the role of neural guidance molecules such as semaphoring-Sema3A- and its receptors.

Methods and results are almost clear but since several passages and data are reported it is difficult to follow read the manuscript somewhere.  

The Authors need to deal with some major critical points before the manuscript may be considered for eventual publication as reported in the attached document.

Reviewer 2 Report

  1. Did the authors identify differences in other Semaphorin family member's differential expression during bone remodeling?
  2. Why there was a significant reduction in Sema3A receptors PLXNA1 and NRP1 during 48h of strain, which was actually opposite to the expression of Sema3A mRNA at the same time point?
  3. Sema3A expression correlates with NRP1 expression during compression but not during strain. Why?
  4. In WB, I do not see any downregulation of Sema3A at 4h as the authors claim. Also in plexinA1 expression in Fig.2B

Both figures in Fig2 B and 2C (a) seem not reliable as beta-actin expression varies a lot in all experimental conditions, which shows there was a problem during the blot development rather than the protein amount. The differences claimed by authors are not clearly seen in western blot. How much protein was loaded during SDS-PAGE? A better representative image has to be presented.

  1. 130, 131 and 132 lines should be shifted to the discussion part. Any justification for this point?

Line 168: ‘were’ should be changed to ‘where’

  1. What are the main differences between strain and compression? Why both kinds of forces have different effects on hPDLF cells? This has to be discussed properly in the discussion.
  2. 2.3 and 2.5 subheading: Semaphorin spelling needs to be corrected. And also here: human semaphoring 3A [10ng/mL]
  3. With so much standard deviation in Fig 5B, I don’t really see an increase in ALPL expression (clearly not significant). With no significance in p-value, how can the authors claim that there is an upregulation of ALPL? Similarly, authors need to clarify how can they claim there is an upregulation of SPP1, BGLAP?
  4. Why the authors have used different concentrations of Sema3A for the experiments (10, 50, 100 and 250 ng/ml) in Figs 5, 6 and 7? With so much variation in the concentration used, the results cannot be compared.